# The Accuracy of Semi-Empirical Quantum Chemistry Methods on Soot Formation Simulation [note 1]

**DOI:** 10.3390/ijms232113371

**Published:** 2022-11-02

**Authors:** Yang Cong, Yu Zhai, Xin Chen, Hui Li

**Affiliations:** 1Institute of Theoretical Chemistry, College of Chemistry, Jilin University, 2519 Jiefang Road, Changchun 130023, China; 2Institute of Systems and Physical Biology, Shenzhen Bay Laboratory, Shenzhen 518055, China

**Keywords:** soot, soot precursors, PAHs, molecular dynamics simulation, semi-empirical, DFT tight-binding, benchmark

## Abstract

Soot molecules are hazardous compounds threatening human health. Computational chemistry provides efficient tools for studying them. However, accurate quantum chemistry calculation is costly for the simulation of large-size soot molecules and high-throughput calculations. Semi-empirical (SE) quantum chemistry methods are optional choices for balancing computational costs. In this work, we validated the performances of several widely used SE methods in the description of soot formation. Our benchmark study focuses on, but is not limited to, the validation of the performances of SE methods on reactive and non-reactive MD trajectory calculations. We also examined the accuracy of SE methods of predicting soot precursor structures and energy profiles along intrinsic reaction coordinate(s) (IRC). Finally, we discussed the spin density predicted by SE methods. The SE methods validated include AM1, PM6, PM7, GFN2-xTB, DFTB2, with or without spin-polarization, and DFTB3. We found that the shape of MD trajectory profiles, the relative energy, and molecular structures predicted by SE methods are qualitatively correct. We suggest that SE methods can be used in massive reaction soot formation event sampling and primary reaction mechanism generation. Yet, they cannot be used to provide quantitatively accurate data, such as thermodynamic and reaction kinetics ones.

## 1. Introduction

Soot, a mass of carbonaceous nanoparticles, is a byproduct from incomplete combustion. The deposition of soot particles in a combustion chamber affects the combustion efficiency and even reduces the life of engines. Atmospheric soot also plays an important role in the global climate system [1]. Moreover, epidemiological evidence has already shown that black carbon, mature soot particles emitted into the atmosphere, is associated with health damage [2,3]. Soot precursor formation processes depend on fuel composition and flame conditions. Some parts of the complicated soot inception mechanisms are still unclear and waiting for more research.

Soot particles are generated from gas-phase species, which are called soot precursors. Although numerous experimental and simulation research studies support the theory that polycyclic aromatic hydrocarbon (PAH) species are major precursors [4,5], the genesis and growth mechanisms of aromatic compounds are still under debate. Various pathways are proposed for the formation of soot precursor, some research studies deem benzene as the first aromatic ring and soot-growing seed [5,6]. Another class of theoretical works suggested that stable cyclopentadienyl radicals formed by C_3_H_3_ and C_2_H_2_ are crucial initial structures to form other aromatic rings [7]. Johansson et al. [8] proposed a soot inception routine, which started from a cyclopentadienyl radical without the direct participation of benzene. Jin et al. [9] also suggested a propargyl radical (CH_2_CCH) and butadiyne (HCCCCH) reaction soot initiation pathway. In the research of PAH growth mechanisms, Frenklach and Mebel [10] proposed a general mechanism named H-activated-carbon-addition (HACA), in which “H-activated” refers to “H-abstraction”, “H-addition”, and “H-migration”, and “carbon-addition” including the addition of acetylene, vinyl acetylene, and even aromatics themselves. Other plausible pathways include phenyl-addition dehydrocyclization (PAC) [11], clustering of hydrocarbons via radical chain reactions (CHRCR) [12], and continuous butadiyne addition to cyclization (CBAC) [12] pathway.

Quantum chemistry calculations are efficient tools used to discover soot formation mechanisms. The thermal data of combustion species as well as a reaction kinetic model can be predicted accurately using post-Hartree—Fock methods and the density functional theory (DFT). Moreover, quantum chemistry can be used in automatic reaction mechanism generation. Nowadays, more research studies use large-scale ab initio molecular dynamics (AIMD) to simulate the complex reactive processes, including soot formation. However, performing a large-scale (regarding size and time) AIMD simulation at the DFT level is still extremely time-consuming. To reduce computation costs, many methods, such as machine learning and fragment-based quantum chemistry, are implemented. Martinez et al. [13] used hardware acceleration techniques to enable large-scale AIMD simulations. Chen et al. [14] simulated a monomethylhydrazine (MMH)/NO_2_ system with 28,650 atoms using a fragment-based AIMD. Alongside the ab initio calculations, reactive force fields and neural network potential energy surfaces (NNPESs) are also widely used to research soot formation [15].

Gas-phase soot formation systems are sparse and full of radicals. These features indicate that fragment-based methods are suitable for soot formation simulation. Yet, even with the fragment-based method and hardware acceleration, a large-scale AIMD at the DFT level or above is still difficult for simulating the soot formation process. Semi-Empirical (SE) quantum chemistry methods may be better choices for balancing the accuracy and efficiency in soot formation simulation. Moreover, SE methods can also be combined with fragment-based methods [16,17,18] and machine-learning techniques [19,20,21,22]. SE methods can also be used in high-throughput calculations, which are necessary for automatic reaction mechanism generation processes. SE methods have already been used in researching the soot formation and carbon particle growing processes. Bai et al. simulated the growth of polycyclic aromatic hydrocarbons from a system with 40 C_2_H_4_ molecules using a DFTB-based ‘nanoreactor’ approach [23]. The Irle and Morokuma groups conducted research on the carbon particle growing process, including soot formation using DFTB [24,25,26,27]. In one of their research studies on fullerene self-assembly in benzene combustion, they provided a DFTB accuracy benchmark about reaction barrier heights and relative energy results in the supporting information [24].

One large SE quantum chemistry method class is based on the Hartree–Fock (HF) theory. This class of methods reduces the computational costs dramatically by neglecting and parameterizing parts of electron integrals. HF-based SE methods have a long history [28,29,30], and have been developed into well-known schemes, such as Austin Model 1 (AM1) [31], parametric method numbers 6 and 7 (PM6 [32], and PM7 [33]). Moreover, these methods are implemented in plenty of quantum chemistry software packages and are widely used [34,35]. AM1 method is a refinement of the modified neglect of diatomic overlap (MNDO) model [36], which improved the description of short-range interaction by adding Gaussian functions to core–core repulsion terms in the Hamiltonian [31]. The PM6 employs diatomic parameters rather than the element-specific parameters in AM1 [32], and it also has *d*-orbital parameters. PM7, the latest parametric method series method, includes intermolecular dispersion and hydrogen bond interaction corrections [33]. Another class of popular SE approach involves density functional tight binding (DFTB) [37,38,39,40,41], which is an approximation of DFT. DFTB is derived from a Taylor expansion of the DFT total energy. According to the order of truncation, DFTB approximations can be classified into DFTB1, DFTB2 (or SCC-DFTB), and DFTB3 models. DFTB models have various extensions, such as DFTB3 with Grimme’s D3 van der Waals corrections (DFTB3-D3). Grimme et al. proposed a family of SE tight-binding models, named GFNn-xTB (n = 0, 1, 2), which focused on molecular properties (geometries, vibrational frequencies, and non-covalent interaction) at the very beginning. The earlier edition, GFN1-xTB [42] was a variant of DFTB3. A few years later, GFN2-xTB [43] was released and it is regarded as a more physically sound method with a less empirical version. GFN2-xTB incorporates the new D4 dispersion model (Grimme et al.).

The accuracies of SE methods (in describing structurally optimized organic compounds) have been validated in various studies. The large amount of twisted soot compounds yielded during MD simulations need further validation. In this paper, we validated the accuracies of several SE methods on soot formation simulation. This study included the benchmark of energies on a set of reactive or non-reactive MD trajectories, energies along intrinsic reaction coordinate(s) (IRC), the geometry of optimized structures, and spin density. The test sets contained soot-relevant compounds with 4 to 24 carbon atoms and different types of reactions, which represent the emergence and early growth of soot precursors. In view of the controversy of the soot formation theory, we selected different types of hydrocarbon compounds to cover more possibilities. Unlike a condensed environment simulation, the interactions between molecules and radicals are weak in a gas-phase combustion reaction system. Because of this feature, the validation of a single molecule or reaction event can represent the whole system.

## 2. Results and Discussion

### 2.1. Similarity of Energy Profiles

A total of 84 MD trajectories from the MD simulations were used to validate the accuracies of seven SE methods (AM1, PM6, PM7, GFN2-xTB, DFTB2, DFTB3) against M06-2x/def2TZVPP-level DFT calculations. For systems with unpaired electrons, spin polarization was also considered in DFTB2. The selected trajectories covered reactive and nonreactive pathways with different molecule sizes. Potential energy along the trajectory was calculated using the SE methods and benchmarked DFT method. The accuracies of the SE methods are represented by the similarities of the potential energy profiles. One of the energy profile comparisons is presented in Figure 1. This trajectory is approximately a 117 fs reaction event of a H_2_CCCCH^•^ radical colliding on a hydrocarbon compound with 20 C atoms. A C_24_H_18_ compound was formed after the formation of the C-C bond. Qualitatively, all six SE methods have similar trends to the energy profile as the benchmark (M06-2x/def2TZVPP). The local maxima and minima of SE methods are close to that of m06-2X. In this part, we will use statistical indicators of maximum unsigned deviation (MAX) and regularized relative RMSE to discuss the similarities of energy profiles. GFN2-xTB has the best performance among all six SE methods, with an RMSE of 13.34 kcal/mol and MAX of 34.98 kcal/mol. DFTB3 yields the second best results (RMSE = 13.51 kcal/mol, MAX = 42.50 kcal/mol), which is slightly better than those of DFTB2 (RMSE = 15.74 kcal/mol, MAX = 51.14 kcal/mol). The energy profiles of PM6 and its updated version, PM7, are close to each other. Moreover, we did not observe an improvement of PM7 over PM6. AM1 performs better than PM6 and PM7 under this circumstance. The other comparisons of energy profiles are shown in Appendix A.

Spearman’s rank correlation coefficient [44] ρ was used to quantitatively measure the similarity of energy profiles. This factor can reflect the similarity of energy profiles and is independent of the selection of reference structures. For two energy profile series, X and Y with *n* points, the coefficient ρ is expressed as
(1)$$\rho =1-\frac{6\sum_{i=1}^{n}d_{i}^{2}}{n(n^{2}-1)},$$
where $d_{i}=R\left(X_{i}\right)-R\left(Y_{i}\right)$ is the difference between the rank (R) of the *i*th potential energy on two energy profiles, and ρ is always in the range from −1 to 1. A larger value of ρ indicates a larger similarity. Moreover, −1 represents the negative correlation (totally opposite) condition. We evaluated the coefficient of the SE methods with the benchmarked DFT method. The results are represented in Appendix A. We used a color scale plot to represent the overall performances of the different SE methods (in Figure 2).

Through the color scale Figure 2, we can intuitively obtain the global performances of all SE methods when calculating the trajectory energy. In this figure, the color of the block is more inclined to green, indicating that the corresponding ρ of the SE method is closer to 1, i.e., the method obtains results closer to those of the DFT reference. In general, GFN2-xTB and DFTB3 have better performances in describing the trajectory energy distribution, while AM1, PM6, and PM7 are obviously not ideal, surprisingly, PM6 and PM7 are even worse than AM1.

The coefficients ρ correspond to the energy profiles calculated by different methods in all MD trajectories. The results show that in all 84 trajectories, the ρ value of GFN2-xTB method exceeds 0.98 (17 times) and exceeds 0.9 (77 times), while ρ of the DFTB3 method exceeds 0.98 (8 times) and exceeds 0.9 (71 times) (see Appendix A), which is in slightly worse agreement with the reference data. Moreover, in the comparison of different methods for the same energy profile, GFN2-xTB obtained the highest ρ (43 times), while DFTB3 obtained the highest ρ (21 times). To conclude, GFN2-xTB is an excellent agreement with the reference DFT data.

In addition, a total of 21 trajectories were adopted to benchmark the method of spin polarization considered DFTB2-SP, of which, the results with spin polarization were 14 trajectories better than those without spin polarization, but after subtracting the ρ scores of DFTB2-SP and DFTB2, the average score difference between the two methods was only 0.00298. Thus, it can be concluded that spin polarization may bring better results for the energy calculation, but the improvement was very limited.

### 2.2. Similarity of Optimized Structures

In addition to the energy calculation, another important aspect is whether the SE methods can give a more accurate optimization of molecule geometry, which is related to the accuracy of the molecular dynamics provided by SE methods. In order to verify the performance of the various methods in the previous section, 10 soot molecules in the MD trajectory were selected as examples. Some of these molecules contained more than 30 carbon atoms, and the largest one contained 50 carbon atoms. This analysis helps to evaluate various SE methods for the performances of complex molecular structures. The data of these 10 soot molecules are provided in Table 1. The molecules above were optimized by the DFT method (M06-2x) to obtain the reference structure, and then the SE methods were used to obtain optimized structures. The RMSEs of the structures optimized by each SE method and the DFT reference structure were calculated; the results are shown in this table.

Table 1 shows that when the GFN2-xTB structures agreed the best with the DFT ones, of which, the RMSDs were the smallest in 4 species among the 10 molecular structures, while DFTB3 had the three smallest RMSDs, AM1 had two, and DFTB2 had one. At first glance, GFN2-xTB still performed the best, but not as prominently as in energy calculations, while PM6 and PM7 performed poorly. Figure 3 presents the largest C_50_H_35_ molecule as an example to show the differences between the structures optimized by various SE methods and the DFT reference structure. The semi-transparent structure in the figure was optimized by the DFT method; if the other structure was exposed less outside the semi-transparent structure, the structure was more accurate. It can be seen from the figure that PM6 and PM7 deviated the most from the DFT reference structure, followed by AM1, indicating that these methods were not ideal for calculating the molecular structure of soot. A complex compound with both polycyclic and flexible hydrocarbon chains, C_50_H_35_, is shown here as an example. In terms of the structure comparison, all SE methods, including PM6 and PM7, can still provide relatively accurate structures. For the large compound with 50 carbon atoms, we noticed that the structural optimization took several days at the DFT level, yet, SE optimization only took minutes.

Furthermore, it can be found that the RMSE values of different molecules optimized by SE methods were not always at the same level; some molecules obtained large RMSE values regardless of the SE method used to optimize them, which means that the molecules were more difficult to obtain accurate structures with SE methods. From the data in Table 1, it is not directly related to the molecular size. For example, the RMSE values of C_15_H_11_ calculated by different methods were significantly lower than the corresponding results of C_34_H_23_. When the SE methods were used to optimize the molecule C_34_H_23_, the obtained structure and the structure optimized by DFT had big deviations, but the RMSEs of the C_47_H_32_ and C_50_H_35_ structures obtained by all SE methods were smaller than the RMSEs of C_34_H_23_. It is worth noting that in the RMSE data corresponding to C_26_H_22_, the RMSE values of AM1, PM6, and PM7 were much larger than those of the other methods, after comparing the structures optimized by the SE method with the DFT reference structures (all structures are illustrated in Appendix A), we found that the difference of the dihedral angle dominated the large RMSE, at the junction between a carbon chain and a bicyclic structure, the dihedral angle change of several atoms led to the overall change of the molecular configuration, which means that compared with the DFTB and GFN2-xTB methods, the AM1, PM6, and PM7 methods may not perform well in optimization calculations for similar structures.

To verify this, we selected a set of 33 structures from the thermochemical database of the reaction mechanism generator (RMG) [45,46]. The 33 structures contained a variety of alkenyl radicals, cyclopentenyl groups, etc., and used the DFT method and the SE methods in Figure 1 to optimize all 33 structures, calculate the RMSEs between the SE method optimized structure and the DFT reference structure (all data are listed in the Appendix A). In general, the DFTB and GFN2-xTB methods can more stably obtain structures that are close to those optimized by the M06-2x. The AM1, PM6, and PM7 methods have large differences between the results of DFT when optimizing the carbon chain structures, but in cases where the main structures of the molecules are cyclic or polycyclic structures, a relatively accurate structure is more likely to be obtained.

### 2.3. Intrinsic Reaction Coordinate Paths

Intrinsic reaction coordinate (IRC) paths were also used to assess the accuracy of these SE methods. The 3 H-migration and 3 C−C formation reactions were considered in this section because of their important roles in the formation and growth of soot. These reaction pathways were adopted from a previous study [47,48,49], where IRC calculations were performed at M06-2x/def2TZVPP level. The reactant, product, and transition state structures were connected by the IRC paths. The six SE methods were used to recalculate the potential energy along the IRC paths, and spin polarization DFTB2 was also considered for systems with unpaired electrons. The energy profiles are presented in Figure 4 with the reactant structures as reference.

Figure 4a,b are isomerization pathways of C_5_H_7_, formed by the reaction of the methylidyne radical with dimethylacetylene. A comprehensive discussion can be found in a previous study [47]. Figure 4a is the IRC of an intramolecular cyclization in which a three-membered ring is formed. The cyclization barrier is 18.36 kcal/mol and the backward barrier is 10.90 kcal/mol for a reference DFT calculation. PM6 and PM7 fail to calculate the barrier because they predict energy minima in the transition state regime. The relative energies of reaction and cyclic products are reversed by GFN2-xTB. AM1, DFTB2SP, and DFTB2 overestimate the cyclization barriers by about 9.49 kcal/mol, 7.82 kcal/mol, and 4.67 kcal/mol, respectively. DFTB3 underestimates the forward reaction barrier by 5.68 kcal/mol. GFN2-xTB largely underestimates the forward reaction barrier by 14.32 kcal/mol. The estimations of the backward reaction barrier height are more accurate than those of the forward reaction. AM1, DFTB2, and spin-polarized DFTB2 have close backward reaction barrier predictions of 10.48 kcal/mol, 9.82 kcal/mol, and 8.78 kcal/mol. DFTB3 and GFN2-xTB underestimate the backward reaction barrier of about 2.96 kcal/mol and 4.00 kcal/mol. In view of these, DFTB2 gives relatively good predictions for this reaction path.

Figure 4b is the IRC that illustrates the intramolecular H-atom migration with a large potential energy barrier height (forward: 47.69 kcal/mol; backward: 65.04 kcal/mol). From the reactant to the product, all seven SE methods can estimate the forward reaction barrier within the error range of −2.83 kcal/mol to 5.29 kcal/mol. The forward reaction barrier prediction of PM7, 47.64 kcal/mol is close to that of the reference DFT result. Spin-polarized DFTB2 overestimates the forward reaction barrier height of about 5.29 kcal/mol, which is the largest error. Moreover, DFTB2 without spin-polarization slightly overestimates the forward reaction barrier height of about 2.05 kcal/mol. GFN2-xTB, DFTB3, PM6, and AM1 underestimate the reaction barrier of about 0.74 kcal/mol, 1.33 kcal/mol, 1.96 kcal/mol, and 2.83 kcal/mol. We also noticed that the backward reaction barrier can be well described by SE methods.

Figure 4c is the IRC of the addition of acetylene to fulvenallene, in which 2 C−C bonds are formed. This IRC profile has a wide transition state area with flat potential energy. The forward reaction barrier is 35.46 kcal/mol and the backward barrier is 92.29 kcal/mol (DFT reference result). All SE methods in validation have difficulties in describing the two-bond formation flat transition state. The DFTB3 method has the best performance among the six SE methods. AM1, PM6, and PM7 overestimate the forward barrier by 7.00 kcal/mol, 3.17 kcal/mol, and 1.32 kcal/mol, respectively. The highest energy structures by these three methods locate at the back part of the TS area. The backward reaction potential energy barrier is close to that of the reference DFT result. However, GFN2-xTB and DFTB2 largely underestimate the forward reaction barrier nearly 20 kcal/mol.

Three isomerization pathways of open-shell E-bridge PAH precursors are illustrated in Figure 4d–f [49,50]. The relevant energies between the reactant and product of pathway (d) are well predicted by all the seven SE methods. Spin-polarized DFTB2 has the best prediction of TS-relevant energy to the reactant. AM1, PM6, and PM7 methods overestimate the barrier height of more than 10 kcal/mol. GFN2-xTB and DFTB2 slightly underestimate the TS energy.

Reaction pathways in Figure 4e illustrate the H atom migration. Surprisingly, all four density-functional tight-binding (DFTB) methods (GFN2-xTB, DFTB2, DFTB3, and spin-polarized DFTB2) fail to describe the energy profile because they largely over-stabilize the transition state structure. The relative energy of TS structures from DFTB2, DFTB3, and spin-polarized DFTB2 are close to or below zero. These faults are caused by fractional-charge errors in which the H atom carries an abnormal amount of positive charge, and induces larger attractive coulomb interaction during the H-transfer. Using the TS structure on DFTB2 energy profile as an example, the nearly isolated H atom has a large positive partial charge of 0.163. Previous studies have located this problem and suggested a solution named constrained DFTB [51,52]. Meanwhile, GFN2-xTB also predicts a very low reaction barrier height. AM1, PM6, and PM7 predict the energy increase of the transition state structure correctly. Yet, PM7 and PM6 largely overestimate the forward reaction barrier.

The reaction pathway in Figure 4f illustrates the intramolecular radical cyclization. A low potential energy five-member ring formed during that reaction. The forward reaction barrier of this reaction from the DFT calculation is 19.72 kcal/mol. Due to the stability of the product, the backward reaction barrier is as large as 41.12 kcal/mol. DFTB series methods (DFTB2, DFTB2-SP, DFTB3, and GFN2-xTB) underestimate the forward reaction barrier. On the contrary, HF-based SE methods (AM1, PM6, and PM7) overestimate the forward reaction barrier. The highest energy structures predicted by DFTB methods locate in front of the TS structure. The TS structure has the largest energy for PM6 and PM7 methods. Yet, the highest energy structure on AM1 energy profile is behind the TS structure.

### 2.4. Spin Density Distribution

Free radical reactions play dominant roles in most soot formation pathways. For examples, a continuous free radical addition reaction is the key component in mechanisms such as HACA, PAC, and CHRCR. For open-shell molecules, the sites with unpaired electrons are generally reactive. Therefore, the locations of free radicals largely affect the reactivity of soot precursors. The spin of unpaired electrons can be considered in HF-based SE methods and spin-polarized DFTB methods. The spin density (a.k.a. magnetization density), which is defined as the spin-up electron density minus the spin-down electron density, $\Delta \rho^{m}\left(r\right)=\rho ↑\left(r\right)-\rho ↓\left(r\right)$, is an effective quantity to characterize the location of a free radical. In this part, we used the large molecule C_50_H_35_ from Section 2.2 as an example to show the spin density predicted by AM1, PM6, PM7, GFN2-xTB, and spin-polarized DFTB2. The spin density from the DFT calculation is also plotted as the reference. The comparison of the spin density is shown in Figure 5.

The spin density from the reference UDFT calculation shows that the unpaired electron spin density is localized at the polycyclic aromatic part of the C_50_H_35_ molecule. Meanwhile, the isosurface also indicates that the unpaired electron is on the π-orbital. Spin-up orbitals are not strictly identical to the corresponding spin-down orbitals in unrestricted SCF calculations. This feature causes the separation between spin-up and spin-down densities. AM1, PM6, and PM7 largely exaggerate the separation of spin-up and spin-down densities. Spin densities also incorrectly appear outside of the polycyclic part in the prediction of PM7. The unwanted spin-up and spin-down separation is weak in the predictions of GFN2-xTB and spin-polarized DFTB2. However, the spin-up electron density from the DFTB2-SP calculation is delocalized to the whole molecule. GFN2-xTB predicts the location of the unpaired electron density accurately compared with the other four SE methods. In summary, the characterization of free radicals on soot molecules is difficult for SE methods, except GFN2-xTB.

## 3. Materials and Methods

### 3.1. SE Calculations

In this work, we compared six semi-empirical methods—AM1, PM6, PM7, GFN2-xTB, DFTB2, and DFTB3. These easy-access SE methods have already been used in various carbonous systems simulations, including soot formation. AM1, PM6, and PM7 calculations, which were performed in Gaussian 16 Rev B.01 (Gaussian, Inc., Wallingford, CT, USA) [53]. AM1 method in Gaussian16 has been modified to use standard integral infrastructures, which are different from the original algorithm in MOPAC [54]; we used the modified version of PM7 for the continuity of potential energy surfaces [55].

GFN2-xTB calculations were performed with xTB 6.4.1 [43], and DFTB2, DFTB2-SP (spin polarization considered) and DFTB3 calculations were performed with DFTB+ release 21.2 [56]. For DFTB2 calculations, a parameter set emphasizing organic molecules, mio-1-1 [57], was used. Parameter set 3ob-3-1 [58] was used for DFTB3 calculations.

### 3.2. MD Simulations

MD simulations were performed to collect reaction events. The soot formation systems were simulated by using fragment-based AIMD and ReaxFF (reactive force field) potential. Both simulations were initialized in the same configurations with 1400 H_2_ molecules, 500 O_2_ molecules, 200 C_2_H_3_ radicals, 1400 C_2_H radical, 100 C_3_H_5_ radicals, 100 CH_3_CHCH radicals, 300 CH_3_ radicals, and 200 H radicals. Large amounts of radicals were introduced for sampling reaction trajectories efficiently. Both simulations used a time step of 0.1 fs and periodic boundary conditions. A fragment-based AIMD simulation at the PW91/6-31+G(d) level was performed using CARNOT [14] with constant particle number, volume, and energy (NVE) at an initial temperature of 1500 K and pressure of 9 atm. The ReaxFF MD simulation was carried out with a constant particle number, volume, and temperature (NVT), and had the same initial conditions as a fragment-based AIMD simulation. ReaxFF MD simulation was performed using the reax/c package in LAMMPS [59].

### 3.3. Selection of the Benchmark DFT Method

DFT is a well-benchmarked method that can simulate the thermodynamic properties of hydrocarbon compounds accurately [60]. In this research, DFT calculations were chosen as benchmarks for validating the performances of SE methods. In order to choose a proper ’functional’ as the benchmark, we compared the performances of several functional (B3LYP [61,62,63,64], X3LYP [65], ωB97XD [66], PW91 [67], PBE [68], PBE0 [69], B2PLYPD3 [70] and M06-2x [60]), highly accurate CCSD(T) calculations. Reactive trajectories HCC^•^ + HCCCH=CH_2_ → ^•^HC=CH−CC−CH=CH_2_ were extracted from the full system simulation for this validation. Potential energies were calculated along this collision trajectory by using the eight density functionals mentioned above and the UCCSD(T) [71] method. def2TZVPP [72,73] was used as the basis set for all QM calculations. All QM calculations in this subsection were performed using the Gaussian16 [53] package. Figure 6 presents the results of the comparison. First, the structure from this trajectory was set to reference.

This 81.5 fs reactive trajectory with a time step of 0.5 fs contained two elementary reactions, H-abstraction and free-radical addition reaction. M06-2x and ωB97XD have the best performances among these eight density functionals and the root-mean-square error (RMSE) of M06-2x was 2.29 kcal/mol, which was slightly better than ωB97XD with an RMSE of 2.88 kcal/mol. In view of that, we chose M06-2x as the QM benchmark for validating other SE methods.

## 4. Conclusions

In this work, we performed a systematic benchmark analysis for assessing the performances of several widely used SE methods (AM1, PM6, PM7, DFTB2, DFTB2-SP, and DFTB3) in describing the properties of soot precursor molecules and their reactions. The predictions from SE methods were compared with the reference DFT calculation (M06-2x/def2-TZVPP). We validated the accuracies of SE methods in various aspects including the MD trajectory energy profile, structural optimization, IRC energy profile, and spin density. The following conclusions were obtained:(1)The energy profiles calculated by SE methods are generally similar to those of reference DFT calculations. The GFN2-xTB method has significant advantages over other SE methods in describing the energy profile of MD trajectories. Compared with HF-based methods (AM1, PM6, and PM7), DFTB series methods have better performances. We also find that the consideration of spin polarization does not significantly improve the performance of DFBT2.(2)SE methods can optimize the structures of the soot precursor well. DFTB methods generally perform better than HF-based SE methods for both small soot-relevant test sets and sets with larger structures. The structures of cyclic or polycyclic rigid molecules are described well using SE methods. Yet, HF-based methods have larger offsets relative to the reference for molecules with flexible side chains.(3)In the validation of IRC energy profiles, SE methods can give accurate relative energies, which can be used for the qualitative analysis of reaction paths in most cases, but are still not suitable for quantitative calculations. In addition, in the case of calculating H-transfer in gaseous PAH precursors, both GFN2-xTB and DFTB methods were observed to underestimate the transition state energies because of the fractional charge error.

In the spin density calculations, AM1, PM6, and PM7 exaggerate the separation of spin-up and spin-down densities, while the DFTB prediction delocalizes the spin-electron density to the entire molecule. They are wrong compared to the DFT reference. Among these SE methods, only GFN2-xTB accurately predicts the position of the unpaired electron density, which can be used to characterize free radicals on soot molecules.

## Figures and Tables

**Figure 1 ijms-23-13371-f001:**
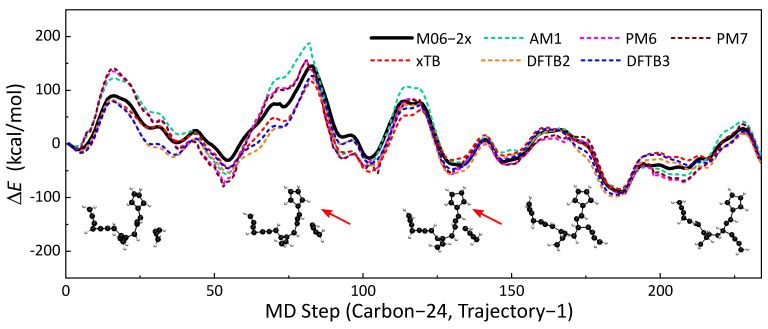
The no. 1 soot trajectory containing 24 carbon atoms was taken as an example; the profiles of ΔE with MD times obtained by different semi-empirical methods (dotted line) are shown in this figure, and the result of the M06-2x (black line) was used as a reference. Each time step of the molecular dynamics took 0.5 fs.

**Figure 2 ijms-23-13371-f002:**
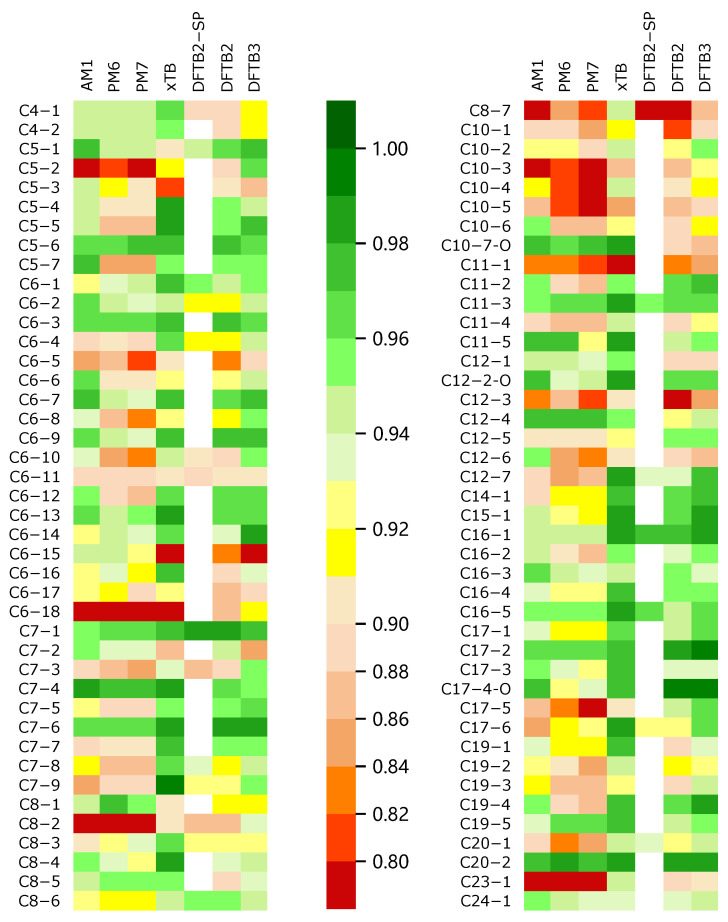
All soot trajectories were calculated by different semi-empirical methods for the energy profiles and the energy profiles were plotted. The DFT method (M06-2x) was used as a reference to score all the profiles; the result of M06-2x had a score of 1.00; a method with a score closer to 1.00 when calculating the energy change was relatively superior; the color appears to be a darker green, and the opposite is closer to red.

**Figure 3 ijms-23-13371-f003:**
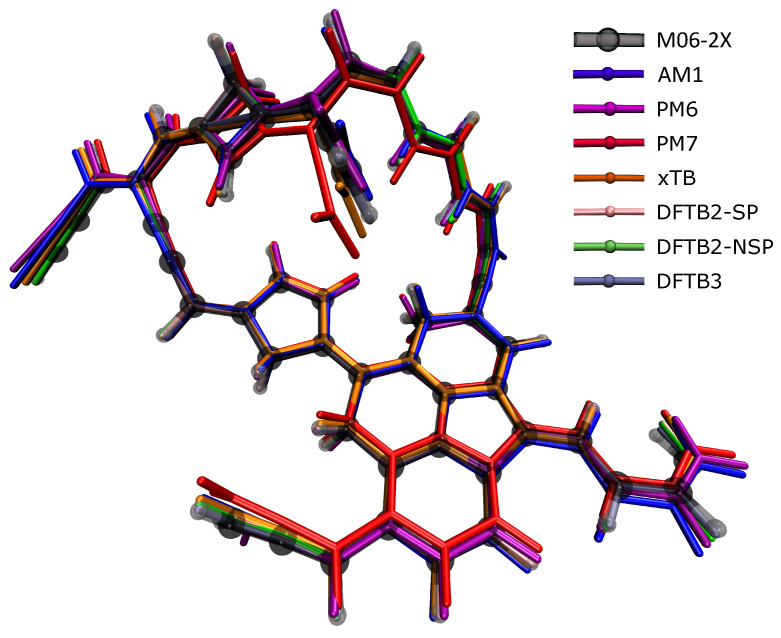
The soot structure with 50 carbon atoms in the figure was optimized using different calculation methods. The calculation result of M06-2x (translucent structure) was used as a reference. Other methods are represented by different colors. It is to be judged whether the method is suitable for optimizing the molecular structure of soot.

**Figure 4 ijms-23-13371-f004:**
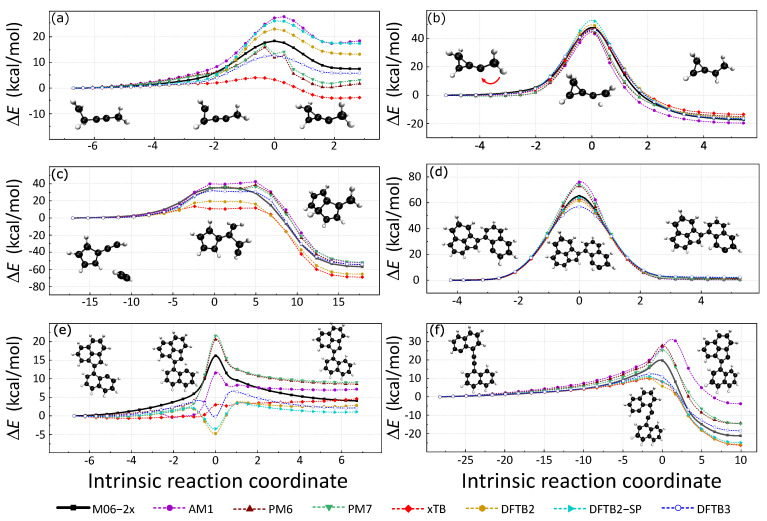
Three C−C formations and three H-migration reactions selected from previous works are shown in this figure. First, we performed the above 6 IRC calculations at the M06-2x/def2TZVPP level, the corresponding data are the black solid lines in the figure. The potential energy corresponding to each data point on the reaction path was then recalculated using various SE methods; each method is represented by a dashed line in a different color.

**Figure 5 ijms-23-13371-f005:**
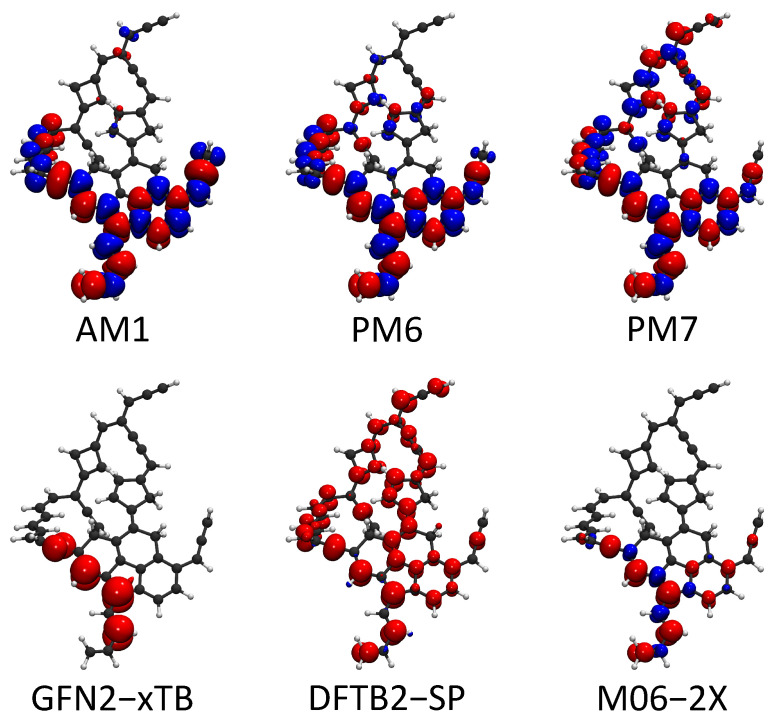
Spin density isosurfaces for the C_50_H_35_. The red color denotes the excess spin-up electron density $\rho ↑\left(r\right)$ and blue denotes the excess of the spin-down electron density $\rho ↓\left(r\right)$. Moreover, the isosurface value is set at 0.003 $e/{\overset{˚}{A}}^{3}$ for both spin-up and spin-down densities.

**Figure 6 ijms-23-13371-f006:**
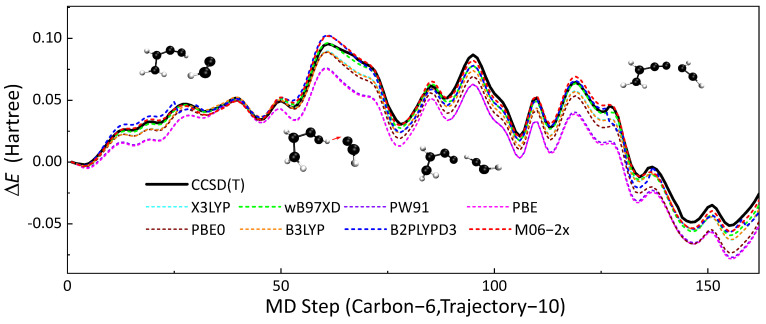
The no. 10 soot trajectory containing six carbon atoms was taken as an example, the profiles of ΔE with MD times obtained by different DFT methods (dotted line) are shown in this figure, and the results of CCSD(T) (black line) were used as the reference. Each time step of molecular dynamics took 0.5 fs.

**Table 1 ijms-23-13371-t001:** The 10 soot molecules were extracted from molecular dynamic trajectories, which were optimized using the M06-2x method as a reference structure, and the structures were optimized separately using various SE methods in the table after rotating and overlapping the coordinates using the Kabsch algorithm and calculating the RMSE of the structure optimized by each method and the reference structure. The smaller the RMSE value, the more similar the structure was optimized to the reference structure, the minimum value of this row is marked as bold.

	AM1	PM6	PM7	GFN2-xTB	DFTB2-SP	DFTB2	DFTB3
C_15_H_11_	**0.0456**	0.0915	0.3517	0.0563	0.0602	0.0602	0.0536
C_18_H_14_	0.0824	0.1040	0.0943	**0.0674**	-	0.0887	0.0761
C_21_H_17_	0.5073	0.5068	0.5143	**0.1463**	0.1689	0.2087	0.2087
C_23_H_15_O_1_	0.1819	0.1710	0.5080	**0.0497**	0.0642	0.0609	0.0656
C_26_H_19_	0.3568	0.6439	0.6862	**0.1690**	0.1931	0.1929	0.2592
C_26_H_22_	1.2711	2.4898	2.5142	0.9765	-	0.7924	**0.7377**
C_28_H_18_	**0.2743**	0.2878	0.2850	0.3956	0.3964	0.3883	0.3957
C_34_H_23_	0.7308	0.8423	0.8707	0.2865	0.2803	0.2870	**0.2595**
C_47_H_32_	0.3241	0.5465	0.6677	0.4234	-	**0.0998**	0.1037
C_50_H_35_	0.3511	0.4144	0.5725	0.2471	0.1177	0.1183	**0.0671**

## Data Availability

Not applicable.

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
