# Peer review of "The Accuracy of Semi-Empirical Quantum Chemistry Methods on Soot Formation Simulation†"

_ijms, 2022, doi:10.3390/ijms232113371_

Round 1

Reviewer 1 Report

The paper "The Accuracy of Semi-empirical Quantum Chemistry Methods
on Soot Formation Simulation" by Yang Cong et. al. reports on benchmark calculations to evaluate the performance of different semi-empirical quantum chemistry methods for the simulation of soot particles.

The scientific topic and the methods applied are well-founded and the results obtained by these calculations are of broad interest, not only for simulating these class of molecules but also for a general judgement of the performance of the semi-empirical methods presented in this work.

Unfortunately, the way the paper is written is partially incomprehensible, full of grammatical mistakes. Just to mention a few examples (how it should be):

Line 43: Quantum chemistry calculations are...

Line 62: .... SE methods can also be combined with

Line 68: The Irle and Morokuma groups...

Line 70: in benzene combustion...

Line 72: the supporting information.

Lines 94-95: GFN2-xTB incorporates the new D4 dispersion...

Line 104: In this work (the first three words have to be deleted).

and so on and so forth.

I strongly recommend to critically review the manuscript by a native speaker or somebody with an extensive knowledge of English.

Two paragraphs (starting at line 191 and starting at line 301) are incomprehensible and should be completely rephrased.

The key inlets of Figure 5 are too small an cannot be read. Figure 6 contains a misspelled method: It's not xTB-GFN2 but GFN2-xTB (at least the latter is used throughout the text).

The references contain some minor mistakes: Most of the journal titles are given in full form, some are abbreviated. This should be unified.

One small scientific problem: For equation (1) not all the ingredients are defined. What is n, what are the lower and upper bounds of the sum, where does the factor 6 in front of the sum come from?

And why should the resulting value be between -1 and 1? We have 1 - something, the latter is strictly positive. This puts an upper bound to 1, but why should there be a lower bound at -1? This should be explained in the text.

Author Response

Response 1: We appreciate for your careful read and spotting the gramma mistakes in our manuscript. The mistakes listed in the review report are all reversed. Meanwhile, we also checked the manuscript thoroughly. Some sentences with grammar mistakes are reversed and highlighted in the reversed edition.

The two paragraphs (starting at line 191 and starting at line 301) mentioned in the review report are fully reorganized.

Response 2: Font size and molecular structures in Figure 5 are enlarged. The misspelling of “GFN2-xTB” is also corrected.

Response 3: All the references are checked and corrected carefully.

Response 4: The equation (1) is the final result of Spearman's correlation factor. The n is the total number of points on energy profiles. The summation is over all n points on energy profile. The equation (1) and description are also reversed according to the review report for clarification. And the number 6 in equation (1) is a derived number rather than a factor. The factor, -1, represents the negative correlation (totally opposite) between two time series. This explanation is also added to the main text.

Reviewer 2 Report

Comments are uploaded.

Author Response

Thank you for your constructive questions and advice.

Response 1: According to you suggestion, we added introduction for elucidate why we selected very different hydrocarbon compounds to our test set. The following sentence is added to introduction (near line 104)

“In view of the controversy in soot formation theory, we selected different types of hydrocarbon compounds as more as possible for covering more possibility”.

Response 2: The type of compounds in figures 4 and 5 are discussed. In the real process of soot formation, the structure of hydrocarbon compounds is extremely complex. Take the compound with 50 carbons in figure 4 as an example, it has both polycyclic and carbon chain parts.

Response 3: For computation time of different methods, we provide some qualitative discussion in the paragraph near line (218-230),

“For the large compound with 50 carbon atoms, we noticed that the structural optimization took several days at DFT level, however, semiempirical optimization only took minutes.”